# Schizophrenia and Alarmins

**DOI:** 10.3390/medicina58060694

**Published:** 2022-05-24

**Authors:** Huan Ma, Ning Cheng, Caiyi Zhang

**Affiliations:** 1Department of Psychiatry, First Clinical College, Xuzhou Medical University, Xuzhou 221000, China; 300102120346@stu.xzhmu.edu.cn (H.M.); 300102120347@stu.xzhmu.edu.cn (N.C.); 2Department of Psychiatry, The Affiliated Xuzhou Oriental Hospital of Xuzhou Medical University, Xuzhou 221000, China; 3Department of Medical Psychology, Second Clinical College, Xuzhou Medical University, Xuzhou 221000, China

**Keywords:** schizophrenia, biomarkers, alarmins, HMGB1, interleukin-33

## Abstract

Schizophrenia, consisting of a group of severe psychiatric disorders with a complex etiology, is a leading cause of disability globally. Due to the lack of objective indicators, accurate diagnosis and selection of effective treatments for schizophrenia remain challenging. The association between schizophrenia and alarmins levels has been proposed for many years, but without solid evidence. Alarmins are prestored molecules that do not require processing and can be released upon cell death or damage, making them an ideal candidate for an early initiator of inflammation. Immunological biomarkers seem to be related to disease progression and treatment effectiveness. Several studies suggest strong associations among the high-mobility group box 1 protein (HMGB1), interleukin-1α, interleukin-33, S100B, heat-shock proteins, and uric acid with schizophrenic disorders. The purpose of this review is to discuss the evidence of central and peripheral immune findings in schizophrenia, their potential causes, and the effects of immunomodulatory therapies on symptoms and outline potential applications of these markers in managing the illness. Although there are currently no effective markers for diagnosing or predicting treatment effects in patients with schizophrenia, we believe that screening immune-inflammatory biomarkers that are closely related to the pathological mechanism of schizophrenia can be used for early clinical identification, diagnosis, and treatment of schizophrenia, which may lead to more effective treatment options for people with schizophrenia.

## 1. Introduction

Schizophrenia is a severe mental disorder characterized by positive, negative, and cognitive symptoms. These symptoms included delusions, hallucinations, amotivation, social withdrawal, deficits in working memory, and cognitive flexibility [1]. Some studies have shown that it affects about 1% of the global population, and accounts for significant healthcare costs. Furthermore, at least 13–15 years of potential life are lost in schizophrenia patients, especially in men [2]. Due to the high complexity of the structure and functional activities of the human brain, the exact etiology and influencing factors of schizophrenia are currently still poorly understood. Multiple factors jointly are likely to contribute to the development of schizophrenia. For decades, a famous hypothesis has been that the dopaminergic and glutaminergic systems are involved in the pathophysiology of schizophrenia [3].

Nevertheless, these hypotheses cannot fully explain the onset, course, and remission of schizophrenia. A growing and compelling body of evidence implicates immunologic dysfunction as the key element in its pathomechanism. In addition, there is a link between inflammatory diseases and schizophrenia. Recently, there is increasing evidence that infection and immune abnormalities are closely associated with the occurrence and development of schizophrenia, and anti-inflammatory drugs can improve some of the symptoms of schizophrenia [4,5,6,7]. The concept of Alarmins was first proposed by Oppenheim in 2005 [8]. It is a structurally diverse group of multifunctional host proteins that are rapidly released due to degranulation, sterile cell/tissue damage or death, or in response to immune induction [8,9]. Most importantly, Alarmins can both recruit and activate antigen-presenting cells, enhancing the adaptive immune response and serving as early warning signals to activate the innate and adaptive immune systems [8,10]. However, the excessive release of alarmins after cell damage can massively amplify the immune response and cause further damage in a proinflammatory loop [11].

Currently, based on their origin, alarmins can be classified into the following three categories [10]: (1) Defensins (α, β), Cathelicidin (LL37/cathelin-related antimicrobial peptide), eosinophil-derived neurotoxin (EDN), granule-derived granulysin; (2) the high-mobility group box 1 protein (HMGB1), high-mobility group nucleosome-binding domain 1 protein (HMGN1), interleukin-1α (IL-1α), and interleukin-33 (IL-33) derived from the nucleus; as well as (3) heat-shock proteins (HSPs), S100 proteins, adenosine-triphosphate (ATP), and uric acid crystals originated from cytoplasm. Alarmins can be used as a molecular biomarker and they are closely related to the degree of tissue damage and the inflammatory process. They can be used to assist clinical diagnosis, assess the seriousness of the disease, prompt clinical intervention in time, reduce morbidity, and help monitor efficacy and prognosis.

In this review, we will focus on the changes and the potential functions of these alarmins in patients with schizophrenia. We hope to determine whether the alarmins can be used as a biomarker of the neuroimmune and neuroinflammation process in the disease of schizophrenia.

## 2. Schizophrenia and Alarmins

### 2.1. Schizophrenia and HMGB1

The HMGB1 gene encodes a protein of 216 amino acids and is composed of three domains: two homologous DNA-binding domains, A and B boxes, and a negatively charged C terminus. The cytokine-activity-region gene of HMGB1 is mainly located in the B box, where the first 40 peptides can induce the production of cytokines such as TNF-α and IL-6, which are highly conserved functional domains that cause an inflammatory response [12,13]. On the contrary, A box has the effect of antagonizing HMGB1 to promote inflammation [14,15]. The C-terminal can combine with the receptor for advanced glycation end products (RAGE) to exert its effect.

HMGB1 is an evolutionarily highly conserved, ubiquitous protein that exists in the nuclei and cytoplasm of nearly all cell types [16]. The biological activity depends on its position in the cell. In the nucleus, HMGB1 acts as a DNA molecular chaperone, which participates in DNA replication, transcription, recombination, and repair. However, when the cell is stimulated by inflammation or trauma, HMGB1 transfers to the outside of the nucleus as damage-associated molecular patterns (DAMPs). In addition to stimulating cells directly, HMGB1 can form immunostimulatory complexes with cytokines and other endogenous and exogenous factors, regulating innate/acquired immunity and extracellular signal transduction [17]. 

HMGB1 interacts with pattern-recognition receptors (PRRs) including receptors for RAGE and Toll-like receptors (TLR). RAGE is a newly identified member of the immunoglobulin superfamily. It can bind to many kinds of ligands and is widely expressed in many kinds of cells such as mononuclear phagocytes, tissue macrophages, cardiac myocytes, lungs, fibroblasts, epithelial cells, endothelial cells, neurons, and smooth-muscle cells [18]. HMGB1 and RAGE can combine to induce the production of cytokines and chemokines through the MAPK pathway, and then the cytokines and chemokines participate in the maturation, migration, and surface-receptor expression of immune cells, leading to related tissue damage and ultimately triggering the inflammatory response [19]. TLR is a natural pattern-recognition receptor in innate immunity. It expresses on the surface of monocytes/macrophages, epithelial cells, and dendritic cells, and plays an important defensive role in the immune response. When HMGB1 is combined with TLR, it will activate NF-κB and then release a large number of inflammatory factors such as IL-6 and TNF, and promotes inflammation [20].

In the central nervous system (CNS), HMGB1 is widely present in neurons, microglia, and astrocytes. It also has a dual role in neurodevelopment and neurodegeneration. On the one hand, HMGB1 promotes nerve development, the growth of nerve axons, and cell migration [15]. On the other hand, HMGB1 is one of the important risk factors in memory-impairment-related diseases, neurodegenerative diseases, and neuroinflammatory-related diseases [21]. Extracellular HMGB1 may act as a proinflammatory cytokine to activate microglia and stimulate the release of other cytokines, aggravating neuronal cell apoptosis. It has been confirmed that anti-HMGB1 treatment has a significant protective effect on neuroinflammation, brain damage, and cognitive impairment in multiple animal models [22,23].

Recent studies have shown that HMGB1 plays an important role in schizophrenia. Related studies on serum HMGB1 levels in schizophrenia patients have obtained a relatively consistent result. Serum HMGB1 levels were significantly higher than those in healthy subjects [24,25,26]. After antipsychotic treatment, the serum level of HMGB1 changed. After treatment, the level of serum HMGB1 decreased compared with that before treatment, but it was still higher than that in the healthy control group [26]. Decreased HMGB1 after antipsychotic treatment may be a marker of psychiatric symptom relief, although it cannot be ruled out that antipsychotics may affect the in vivo synthesis of the studied alarmin [27]. The expression level of serum HMGB1 differs between genders. The serum level of HMGB1 is significantly higher in male patients with schizophrenia than in healthy subjects, which does not depend on the episode and treatment [24]. It suggested that an inflammatory response characterized by elevated HMGB1 levels could persist in the acute or chronic course of schizophrenia.

Moreover, there are some associations between serum HMGB1 levels and clinical symptoms in patients with schizophrenia. Studies have shown that Fibromyalgia and Chronic Fatigue Syndrome Score (FF) are positively correlated with HMGB1 in patients with schizophrenia [28]. In addition, both the Hamilton Depression Rating Scale (HDRS) and the Hamilton Anxiety Rating Scale (HAM-A) scores were strongly positively correlated with HMGB1 serum levels [29]. An interesting finding showed that an increased level of HMGB1 highly significantly predicted PHEM (psychosis, hostility, excitation, and mannerism), negative symptoms, FTD (formal thought disorders), executive function, working memory, and attention [25]. However, there was no significant correlation between HMGB1 and the Positive and Negative Syndrome Scale (PANSS) score or the Calgary Depression Scale for Schizophrenia (CDSS) score [24]. All the studies suggested that effective symptoms, somatic symptoms, negative symptoms, and neurocognitive function of schizophrenia are related to immune-inflammatory response.

Currently, we do not know whether changes in serum HMGB1 before and after antipsychotic treatment can be used to assess treatment response, and the exact dynamics of schizophrenia remain unclear. At present, there are still few studies in this area, and the sample size is small. More studies are needed to investigate the functions of HMGB1 in patients with schizophrenia.

### 2.2. Schizophrenia and IL-1α

The gene encoding human IL-1α is located on the long arm q13-21 of chromosome 2. IL-1α functions as a proinflammatory cytokine, not only by binding to the receptors on the cell’s surface but also by transporting it to the nucleus [9]. It is enriched in activated microglia, macrophages, and lymphocytes. During the necrotic and apoptotic cell death, the IL-1α precursor is completely released and triggers the IL-1 receptor on tissue macrophages to promote the activation of phagocytes, resulting in distinct types of inflammatory responses, including sterile inflammation [9,30]. IL-1α also can activate mast cells and basophils to induce mast-cell maturation and the release of inflammatory mediators, including TNF-α, IL-1β, and IL-4 [31].

The cytokine IL-1α may play a role in the development and maturation of the nervous system. It has been found that IL-1 is expressed both in the brain parenchyma during embryonic neurodevelopment and in the adult brain [32]. The previous study indicated that IL-1α released by microglia affected neural degeneration, and neural protection participated in the regulation of synaptic plasticity. Moreover, IL-1α has a nutritional effect on host dopaminergic neurons in vivo and in vitro, which is closely related to the pathophysiology of schizophrenia [33,34,35].

At the gene level, data from genetic association studies are still inconsistent. Katila et al. pointed out that IL-1a (−889) allele 2 in schizophrenia was somewhat higher as compared to the controls. They proposed that the cytokine aberrations in schizophrenia are, to some degree, genetically determined [36]. Srinivas et al. provide evidence of strong independent associations of the IL1A gene with schizophrenic patients in the Dravidian South Indian population [37]. However, studies in Japanese do not support genetically determined changes in the IL-1 gene complex that increase susceptibility to schizophrenia [38].

Previous findings showed that there is contradicting evidence (elevation, decrease, or no change) regarding levels of proinflammatory cytokine IL-1a in plasma and cerebral spinal fluid of patients with schizophrenia [39,40,41]. These changes seem to be based on the stage of the disease dependent. A study showed that the level of IL-1 was no significant change after treatment with drugs or modified electroconvulsive therapy (MECT) [42]. Furthermore, the variation of IL-1 did not correlate with the reduction rate of the PANSS. However, an elevated level of IL-1α was found in patients with schizophrenia with metabolic syndrome even after 6 weeks of treatment with atypical antipsychotics [43]. The decreased expression of IL1-α and interferon-gamma-inducible protein (IP)-10 was found in patients with schizophrenia [44]. As mentioned, they conjectured that there was a suppressed common upstream pathway.

Among available therapies, three biologics can reduce the activities of IL-1a: the monoclonal antibody bermekimab, the IL-1 soluble receptor rilonacept, and the IL-1 receptor antagonist anakinra [45]. Nonetheless, no relevant research has been found on the inhibitory treatment of IL-1a in schizophrenia. Further research is needed to evaluate the association between the IL-1a gene and the risk of schizophrenia, as well as their correlations in different populations. Moreover, we need to explore the distribution of IL-1 in different brain regions and its pathological mechanism. It will be helpful for treatment in a targeted way.

### 2.3. Schizophrenia and IL-33

IL-33′s mRNA (2.7 kb) encodes a protein of 270 residues and is expression in the human brain [46]. IL-33, a member of the IL-1 family [47], has a three-dimensional structure with 12 β-strands arranged in a β-trefoil fold. IL-33 is mainly secreted by astrocytes and endothelial cells [48,49,50]. Whether oligodendrocytes [46,49] and neurons [51] can secrete remains controversial. IL-33 is present in a large amount in the nucleus of the cell, while a small amount is present in the cytoplasm. It is recognized as a key role in contributing to tissue homeostasis and response to environmental stresses by innate and adaptive immunity.

Some evidence showed that IL-33 may act as both a cytokine and a nuclear factor [52]. Upon release, IL-33 binds to its receptor complex comprising ST2 and IL-1RAcP (IL-1 receptor accessory protein) to activate the nuclear factor-κB and mitogen-activated protein kinase pathways, thereby driving multiple biological processes [53,54]. ST2 receptor is expressed on mast cells and other immune cells, including dendritic cells, monocytes, granulocytes, and Th2 lymphocytes [10,11]. IL-33 and its receptor ST2 participate in the interaction between immune cells and play an important role in diseases of the central nervous system.

IL-33 is indispensable for maintaining the promotive and suppressive function of Treg cell numbers in injured tissues [55]. On one hand, the IL-33/ST2 axis promotes CD4^+^ T-cell expansion and conversion of CD4^+^ Foxp3^−^ T cells into CD4^+^ Foxp3^+^ Treg cells in the periphery [56]. Treg cells can prevent astrogliosis and the ensuing neuronal apoptosis via amphiregulin secretion [57]. On the other hand, increased proinflammatory response and astroglial overdrive and microglial pruning may reduce the number of Treg cells in schizophrenia [58].

Studies on serum levels of IL-33 in patients with schizophrenia and healthy individuals have not yielded consistent results. A study showed that the serum levels of IL-33 and sST2 were similar between patients with schizophrenia and controls, but there was no significant statistical difference [59]. Another study found that serum levels of IL-33 differed by episode and gender. Some researchers found that the serum levels of IL-33 in first-episode schizophrenia patients were significantly higher than those in relapsing schizophrenia patients and healthy controls [60]. There were significant differences in serum IL-33 and ST2 levels in schizophrenia patients compared with healthy patients, especially in men, but not in women [24].

Before and after antipsychotic treatment, a study found that there were no significant differences in IL-33 plasma levels in antipsychotic-naive patients with schizophrenia [61]. However, this study is not representative. In follow-up studies, we can further expand the sample and extend the time of antipsychotic drugs to explore changes in serum IL-33 levels.

There are also some associations between serum IL-33 levels in patients with schizophrenia and their clinical symptoms. Schizophrenia patients with higher levels of IL-33 and sST2 have better cognitive performance, especially in verbal memory, fluent attention, and processing speed [59]. Some scholars speculate that this may be the regulation of IL-33 on plastic events in the patient’s brain. In chronic schizophrenia, there was no significant correlation between IL-33 and PANSS scores, nor between CDSS scores [24]. However, another study found that serum IL-33 levels were positively correlated with positive symptoms of excitement, suspicion/persecution, and hostility, and with general symptoms of anxiety and nervousness during remission of schizophrenia [60].

In follow-up studies, it will be more interesting if we investigate the downstream targets of IL-33/ST2 signaling and the Treg pathway in schizophrenia.

### 2.4. Schizophrenia and Heat-Shock Proteins (HSPs)

Heat-shock proteins (HSPs) are a group of protein chaperones that can protect cells from stress. They exist in all eukaryotic and prokaryotic cells. HSPs play an important role in the process of neurodevelopment. For example, a study has shown that HSPs may promote or inhibit neurodevelopment through specific pathways regulating cell differentiation, neurite outgrowth, cell migration, or angiogenesis [62]. As a member of the alarmins, HSP60, 70, 90, and 96 act on the TLR2 or TLR4 receptors to maintain the normal physiological function of the cell and the homeostasis of the environment [10].

HSP60 and HSP10 are localized head-to-head on chromosome 2, separated by a bidirectional promoter [63]. HSP60 has the functions of coordinating cell apoptosis, participating in inflammation and immune response. It has dual proinflammatory and anti-inflammatory effects. On the one hand, HSP60 and TLR4 are involved in mediating paraquat-induced microglial inflammatory responses [64]. On the other hand, HSP60 induces NF-κB phosphorylation in vitro and in vivo and increases IL-1β secreted by microglia, thereby participating in innate immunity and exerting anti-inflammatory effects [65]. HSP60 is closely related to the occurrence and development of neuropsychiatric diseases. Some studies indicated that the levels of the human 60 kDa heat-shock protein antibodies increased in patients with schizophrenia [66,67].

HSP70, a multigene family that codes for numerous 70-kDa proteins, has been linked to the pathogenesis and therapy of schizophrenia. HSP70 delivered by exosomes plays a neuroprotective role through multiple mechanisms [62]. The most studied members of HSP70s include HSP70-1 (HSPA1A), HSP70-hom (HSPA1L), and HSP70-2 (HSPA1B) [63]. Among them, HSPA5 and HSPA1B have recently been shown to be potentially associated with the occurrence of anxiety, mood disorders, and schizophrenia. Recently, it has been proposed that HSPA1B polymorphism may be associated with schizophrenia and suicidal behavior of schizophrenic patients [68].

In animal experiments, studies have shown that haloperidol-induced dopamine hypersensitivity increases the level of HSP-70 in the hippocampal CA-3 region and nucleus accumbens of rats [69]. Another animal experiment found that maternal isolation enhanced HSPA1B mRNA and proteins expression in the blood and medial prefrontal cortex (mPFC) of juvenile and prepubertal rats [70]. This increase was accompanied by increases in HSPA1A/1B protein levels in the mPFC and hippocampus of young rats and persisted into adulthood. This study suggests that maternal separation may produce persistent overexpression of HSPA1B and HSPA5 in the brain and blood, which may further lead to changes in brain function. Clinical studies have found that patients with schizophrenia or related psychosis have increased antibody reactivity to Hsp70 and Hsp90AB1 proteins in cerebrospinal fluid (CSF) and serum [71]. It suggested an ongoing autoimmune-mediated process in the central nervous system of patients with schizophrenia. However, the sample size of this study was small, and further samples are needed to verify the results.

HSP90 is an essential molecular chaperone in the cell responsible for the stabilization, maturation, and activation of many client proteins [72], and typically acts downstream of HSP70. gp96 belongs to the heat-shock protein 90 families, and is a highly conserved and ubiquitous glycoprotein. As an endoplasmic reticulum protein, gp96 plays an important regulatory role in maintaining endoplasmic reticulum homeostasis, endoplasmic reticulum stress, and calcium homeostasis. The pieces of evidence show that antipsychotic drugs, including olanzapine, aripiprazole, and blonanserin, produce similar GABAergic interneurons to NG2(+) neurons/glial precursor cells via increasing the production of intracellular HSP90 in the adult rat brain [73].

However, there are not enough studies on the above-mentioned heat-shock proteins and clinical symptoms of schizophrenia. More research needs to focus on the relationships between heat-shock proteins and schizophrenia.

### 2.5. Schizophrenia and S100 Protein

The S100 protein is an acidic calcium zinc-binding protein found in the bovine brain by Moore in 1965. It is easy to pass through the blood–brain barrier because its molecular weight is small (9–13 ku). S100 protein genes comprise 13 members present as a cluster on chromosome 1q21 [74]. Structurally, the S100B protein consists of two alpha helix-loop-helix calcium-binding proteins involved in cytoskeleton formation and cellular proliferation [75]. The conformation and amino-acid composition of this protein is highly conserved among vertebrate species. It implied that the different structures may have an extremely conserved biological role and are functionally specific.

In the nervous system, S100B is concentrated in astrocytes and other glial-cell types, such as oligodendrocytes, Schwann cells, ependymal cells, retinal Muller cells and enteric glial cells. It has also been reported to be located in definite neuron subpopulations [76]. Increased S100B release mediated by astrocytes and oligodendrocytes may lead to neuroinflammatory processes by the activation of microglial expression of COX-2 and iNOS and cause dysfunction of neurons and apoptosis [77]. Moreover, astrocytes and oligodendrocytes are inseparable from the occurrence and development of schizophrenia.

In the case of neuronal damage and blood–brain barrier damage, S100B can leak into the cerebrospinal fluid and systemic circulation. Its levels in biological fluids (cerebrospinal fluid, peripheral and cord blood, urine, saliva, amniotic fluid) are recognized as a reliable biomarker of active neural distress. S100B protein is abundant in the central nervous system. It functions as a damage-associated molecular-pattern molecule and acts with RAGE and TLR-4 receptors. has higher activity than other members of the S100 family in the brain. A high level of S100B could trigger tissue reaction to damage via its receptor for advanced glycation end-products in a series of different neural disorders. 

S100B has been proposed as a marker of astrocyte activation and brain dysfunction. Preclinical studies and clinical reports suggested that a higher concentration of S100B in the peripheral blood and cerebrospinal fluid was found in patients with schizophrenia when compared with control [78]. Schizophrenia patients have higher S100B concentrations than healthy controls. However, there is no correlation between the general characteristics such as age, body mass index, disease course, age of onset and S100B serum levels in patients with schizophrenia. A previous study showed a diurnal variation in serum S100B concentrations in patients with acute paranoid schizophrenia [79]. Additionally, the results of the study on the sex of schizophrenia and S100 serum levels are still controversial [80,81]. The previous study found that African-American subjects had significantly higher levels of S100B than Caucasian subjects. Thus, ethnicity or race should be given serious consideration when studying and interpreting S100B levels in patients with schizophrenia [80]. 

In clinical studies, there is controversy as to whether the level of S100B is related to the symptoms of schizophrenia. A study considered that there were no significant correlations between plasma S100B and psychotic symptoms or cognition [82]. Another study discovered that cerebrospinal fluid S100B levels showed a positive correlation with PANSS total, positive, and general scores in patients with schizophrenia [83]. Through autopsy, some researchers found that only in the dorsolateral prefrontal area, the S100B level of schizophrenic patients was significantly higher than that of healthy people [84]. This indicates that the functional changes of the S100B protein in the dorsolateral prefrontal cortex are very important to the etiology of schizophrenia. In addition, more oligodendrocytes are secreting S100B protein in the white-matter region of the brain in paranoid schizophrenia than in residual schizophrenia [84].

Compared to mental health controls, S100B in the nuclear proteome of the corpus callosum from schizophrenia patients was weakened via mass spectrometry. The role of S100B in glial and neuronal cells is based on the concentration [85]. The lower concentrations of extracellular S100 beta act as a growth-differentiating factor, while the micromolar concentrations induce apoptosis [86]. Therefore, it is speculated that fewer oligodendrocytes are maturating in schizophrenia brains. After using voxel-based morphometry to detect the white-matter structures as obtained from T1-weighted MR-images and measuring serum S100B levels, some researchers found that there were significantly different correlations between S100B levels and local white-matter formations between the first episode and recurrent episode patients [87]. That is to say, S100B is involved in an ongoing dynamic process associated with local structural changes in the white brain matter of schizophrenic patients.

The dynamic changes of S100B protein levels can still be used as one of the important indicators for early diagnosis of neurological diseases, comprehensive treatment, drug-efficacy evaluation, and disease-outcome prediction. However, which rules exist in the dynamic changes of S100B in the human body are still worthy of further development. Exploring the predictive effect of outcome in different types of schizophrenia is also worth further investigation. The S100A16 protein belongs to the S100 protein family, which is found in abundance in the brain and adipose tissue [88]. Figura et al. recently discovered that the S100A16 protein concentration in Parkinson’s disease patients’ saliva was lower than in healthy controls [89]. This is a fascinating find, because this protein has never been explored in the context of schizophrenia. In the follow up, researchers need to conduct basic research on various types of schizophrenia and S100B. In addition, the role S100B played in the improvement of schizophrenia through psychotropic drugs, electroconvulsive therapy, and transcranial magnetic stimulation also needs to be studied.

### 2.6. Schizophrenia and Uric Acid

Uric acid (UA), a metabolite of human purine nucleotides, is a powerful antioxidant with the function of scavenging singlet oxygen and free radicals [90]. Purine compounds exert neurotransmission and neuromodulatory effects in the brain. UA is the final product of purine metabolism in the human body. An epidemiological survey has shown that premenopausal blood uric-acid levels are significantly lower in women than in men, and postmenopausal blood uric-acid levels are similar in men, suggesting that sex hormones play an important role in maintaining the body’s uric-acid balance. Furthermore, the increased level can affect the activity of other neurotransmitters, such as dopamine, gamma-aminobutyric acid, glutamate, and serotonin, which are involved in the pathophysiological process of schizophrenia.

Studies have shown that blood uric-acid levels in patients with schizophrenia are lower than normal [91,92]. However, the other studies are controversial [93,94]. A meta-analysis [95], including 17 studies of 2027 subjects with schizophrenia, suggested that there were no significant differences in UA levels between schizophrenia subjects and healthy controls. However, other studies found that decreased UA levels may be a potential risk factor for schizophrenia in men and Americans. Regarding the reasons for the decrease in serum uric-acid levels in patients with schizophrenia, some researchers conducted a study on different brain regions of patients with schizophrenia by using the enzyme activity detection method and found that Xanthine oxidase (XO), a kind of enzyme, can oxidize xanthine to uric acid in the human brain [96]. The activity of xanthine oxidase decreased significantly in patients with schizophrenia when compared with healthy.

A study found that the effect of schizophrenia risk on serum uric-acid levels may be causal, while the effect of serum uric-acid levels risk on schizophrenia is unlikely to be causal [97]. This suggests that uric acid may be a useful potential biomarker for monitoring the treatment or diagnosis of schizophrenia, rather than a therapeutic target for schizophrenia.

Until now, the reasons for the changed level of serum uric acid in patients with schizophrenia cannot be determined. Metabolic abnormalities, smoking antipsychotic treatment, and diet may contribute to it. In the future, these factors should be considered as much as possible in the follow-up research design before further exploring the pathological mechanism related to schizophrenia and uric acid. Beyond that, it is unclear whether the uric acid in the cerebrospinal fluid is similar to the uric acid in the blood. Therefore, more research should focus on cerebrospinal fluid UA levels in schizophrenia patients to validate the current findings.

## 3. Conclusions

Taken together, alarmins play an important role in the pathogenesis of schizophrenia. Although no effective biomarkers have been identified for the early clinical diagnosis of schizophrenia and monitoring of efficacy, it has been demonstrated that the central and peripheral immune-inflammatory state is an important component of the early and late stages of the disease. Alarmins appear to be important markers of disease state in schizophrenia, but the causal relationship between them remains unclear. In the future, it is necessary to further verify the correlation between alarmin and schizophrenia and to further explore the internal mechanism of the occurrence and development of the disease. In addition, serum alarmin levels are a useful marker in the pathology of various neurological diseases, but it is unclear whether alarmin levels in the blood can well reflect central system immune inflammation. Therefore, further studies on the levels of cerebrospinal fluid alarmin in patients with schizophrenia are warranted.

To date, most research on anti-inflammatory drugs has been obtained through in vitro administration in mouse models, and it is unclear to what extent these observations apply to humans. Therefore, further studies with longer follow-up periods in more patients with schizophrenia are needed. Alarmin levels in patients with schizophrenia are associated with obesity, diabetes, circadian-rhythm changes, and seasonal changes. it is advisable to control for these confounding variables while studying. The efficacy of a new adjuvant drug-therapy strategy targeting the immune-inflammatory pathway in relieving the symptoms of schizophrenia will bring new opportunities for drug development if it is clinically proven, and will be more beneficial to the diagnosis and treatment of the disease. Image-guided convection enhanced delivery (CED) technologies are improving, allowing for more efficient direct infusion of medicines and biologics into the brain parenchyma. As a result, it is an appealing drug-delivery approach. Currently, the technology is mostly used in the treatment of brain tumors, Parkinson’s disease, and multiple-system atrophy (MSA) [98]. It is envisaged that this technology will produce fresh discoveries in the spiritual field in the future. 

## Data Availability

Not applicable.

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
