# Peer review of "Schizophrenia and Alarmins"

_medicina, 2022, doi:10.3390/medicina58060694_

Round 1
Reviewer 1 Report
Authors have done a review of alarmins in schizophrenia. The content of the article is in depth with reference to appropriate studies. They have done a great job explaining roles of individual alarmin and their possible implication in Schizophrenia. It is clear that more research is needed on the topic to identify potential biomarkers and treatment options and authors have been able to draft this manuscript to convince the readers for the same backing with relevant research
Author Response
Thank you very much for your careful review and affirmation of our work. This review took a lot of time and effort to write. Current research on schizophrenia and immunological inflammation has made some progress, but it is still insufficient. In the future, our work will focus on the role and possible meaning of alarmins in schizophrenia to determine potential biomarkers and treatment options. We hope that this study will serve as a resource for future research.
Reviewer 2 Report
Mas et al. discussed the evidence of central and peripheral immune findings in schizophrenia, their potential causes, and the effects of immunomodulatory therapies on symptoms and outlined potential applications of these markers in managing the illness. This is very comprehensive review in very common psychiatric disorder. The manuscript is written correctly with good language style. I have several suggestions which can improve the manuscript. Firstly , please discuss potential findings in the context of Parkinson's disease (like S100B, the protein found in patients with Parkinson/s disease is S100-A16, (see Figura et al. Brain Sciences. 2021; 11(5):661. and Fiandaca et al. Neurol Neurochir Pol 2020;54(3):220-231.). Therefore I recommend minor revision.
Author Response
Thank you very much for spending a lot of time and effort making valuable suggestions for our manuscript. We have carefully considered your suggestion, read the relevant literature, and made the following changes:
(1) We reviewed about the recently discovered S100A16 protein in Parkinson's disease and provided references. (page7, line354-358).
(2) We introduced the methods, and technologies for Parkinson’s Disease and cited relevant references. (page9, line419-424).
Reviewer 3 Report
The manuscript is very interesting, although it is not an original work in which the experiment has been shown.
The author presented the impact of several multifunctional proteins (HMGB1, IL-1α, IL-33, heat shock proteins, S100 protein, and uric acid), using extensive literature, almost half of which is from the last five years, which only shows how much is a topic and this, a kind of meta-analysis is important and interesting to nowadays science.
Supporting with professional knowledge is an important basis, but I believe that laboratory work can be considered relevant for encouraging and launching novelties and therapeutic procedures in the treatment of diseases, and even schizophrenia described by the author.
I encourage the authors, and I think that the manuscript should be supported
Author Response
Thank you so much for your encouragement and support, which will help to propel our scientific study forward. Work in the laboratory is both vital and exciting. There is still a lot of laboratory work to be done on the etiological mechanism, clinical diagnosis, and treatment of schizophrenia. In the future, we can do some experiments and meta-analysis according to your suggestions. This review is expected to act as a reference for laboratory research. A vast corpus of research is expected to provide new hope for the treatment of schizophrenia.